# Japanese Diet Indices and Nutrient Density in US Adults: A Cross-Sectional Analysis with NHANES Data

**DOI:** 10.3390/nu16152431

**Published:** 2024-07-26

**Authors:** Marin Aono, Serika Ushio, Yuno Araki, Ririko Ueno, Suzuna Iwano, Aru Takaoka, Yasutake Tomata

**Affiliations:** School of Nutrition and Dietetics, Faculty of Health and Social Services, Kanagawa University of Human Services, 1-10-1, Heisei-cho, Yokosuka 238-8522, Japan

**Keywords:** Japanese diet, Japanese dietary pattern, Japanese Diet Index, nutrient intake, nutrient density, Nutrient-Rich Food Index, 24-hour dietary recall method, cross-sectional study, NHANES, the United States of America

## Abstract

Background: Previous studies have shown that Japanese dietary patterns are associated with high nutrient density. However, these studies were limited to the Japanese population. We examined this association in the US population. Methods: A cross-sectional analysis was conducted using data from the National Health and Nutrition Examination Survey (NHANES) 2017–2018. We included 3138 people aged 20–79 years. Food and nutrient intake data were based on the 24 h recall method. Three Japanese diet indices were used: (1) Japanese Diet Index (JDI, based on 9 food items), (2) modified JDI (mJDI, based on 12 food items), and (3) weighted JDI (wJDI, selected and weighted from mJDI food items). The nutrient density (ND) score was calculated based on the Nutrient-Rich Food Index 9.3. Spearman’s rank correlation coefficients were calculated. Results: The correlation coefficients with the ND score were 0.24 (*p* < 0.001) for the JDI and 0.38 (*p* < 0.001) for the mJDI. The correlation coefficient between the wJDI and ND score was 0.48 (*p* < 0.001). The three Japanese diet indices were correlated with the ND score in all racial groups (*p* < 0.001). Conclusions: Even among the US population, higher degrees of Japanese diet defined by the JDI or mJDI were associated with higher nutrient density.

## 1. Introduction

From an ecological perspective, the dietary characteristics of the Japanese (i.e., the Japanese diet) have been cited as one of the factors that have enabled the Japanese to have the world’s highest class of longevity and healthy life expectancy [1]. For example, regarding nutritional measures for coronary heart disease in the United States, characteristic differences in dietary intake status between the United States and Japan have been noted [1,2]. An epidemiological study of Japanese immigrants also reported higher rates of coronary heart disease in Japanese people living in the United States than in those living in Japan [3,4]. Based on these findings, an attempt to define the Japanese diet score in the United States (Japanese Americans) has also been conducted [5].

Previous Japanese cohort studies have reported that the Japanese diet is associated with better health outcomes, including the risk of all-cause mortality, cardiovascular disease mortality, incident functional disability, and incident dementia [6,7,8,9,10,11]. One possible mechanism for these associations in the Japanese diet is assumed to be a better nutritional balance (e.g., nutrient density) [12,13].

The Japanese Diet Index (JDI) is one indicator of adherence to the Japanese diet, with scores calculated for nine food items (“rice”, “miso soup”, “fish”, “green and yellow vegetables”, “seaweed”, “pickles”, “green tea”, “beef and pork”, and “coffee”) [12,13]. Furthermore, a systematic review of the components of the Japanese diet indicated that three food items (“soybeans and soybean foods”, “fruits”, and “mushrooms”) were also major components in addition to the nine food items of the JDI [14]. Therefore, as a second index regarding the JDI, a previous study defined the modified JDI (mJDI), including these 12 food items [12]. As a third index of the JDI, the weighted JDI (wJDI) has also been defined to identify not only factors for Japanese dietary patterns but also the specific weights that are considered more desirable in terms of nutrient density [12]. The wJDI is defined by food items that are significantly associated with the nutrient density score within the 12 food items of the mJDI and are weighted based on the standardized regression coefficients (i.e., the strength of association between the food items and the nutrient density score) using regression analysis [12].

A previous study showed a positive correlation between three Japanese diet indices (JDI, mJDI, and wJDI) and nutrient density scores among Japanese adults [12]. Specifically, two previous studies have shown a positive correlation between JDI and nutrient density scores among Japanese adults [12,13]. A previous study showed that the mJDI was more strongly associated with nutrient density score than the JDI [12]. Furthermore, one previous study has shown that nutrient density is higher when “green and yellow vegetables”, “soybeans and soybean foods”, and “fruits” are more heavily weighted, and “rice” is consumed in smaller amounts [12]. However, there are few reports on the association between Japanese diet indices and nutrient density. In addition, the previous study included only Japanese adults [12]. To our knowledge, no previous study has examined these indices in non-Japanese populations. It remains unclear whether previous findings can be generalized to non-Japanese areas in terms of external validity. Therefore, there is a lack of evidence that nutritional interventions based on such Japanese diet indices can improve nutrient density in the non-Japanese population. Testing whether the association between dietary indices and nutrient density can be replicated in various non-Japanese populations may provide useful insights into empirical dietary content in other countries, such as the United States.

The purpose of the present study was to examine the association between three Japanese diet indices and nutrient density among populations of various ethnicities in the United States.

## 2. Materials and Methods

### 2.1. Study Design

The present study was based on a cross-sectional analysis of dietary intake data from the National Health and Nutrition Examination Survey (NHANES) 2017–2018. The NHANES is a nationally representative cross-sectional survey designed to monitor the health and nutritional status of the United States population [15,16]. The present study used data from participants who responded to an in-person dietary survey.

### 2.2. Study Population

A flowchart describing the selection of participants is shown in Figure 1. The NHANES target population is the noninstitutionalized civilian resident population of the United States [17,18]. The NHANES 2017–2018 survey included all ages [15,16,17]. Of the 16,211 persons who were eligible to participate in the NHANES 2017–2018 survey, we excluded 7507 people who did not complete the survey, 1064 people who did not provide valid responses to the first day’s dietary survey, 3138 people who were aged <20 years or ≥80 years, 49 people who were pregnant, and 38 people who were lactating (*n* = 4415). Furthermore, among 4415 people, we also excluded 1277 people who were potential outliers in the variables of nutrient intake used in the present analysis (bottom 2.5% and top 2.5%; total of 5%). Since the distribution of energy intake is widely spread from 0 to 12,501 kcal in NHANES 2017–2018 survey data, previous studies have reported the need to consider outliers focusing on numeric values of nutrient intake [19]. Consequently, 3138 individuals were included in the present analysis.

### 2.3. Dietary Survey Data

Dietary survey data were based on the 24 h recall method conducted at the Mobile Examination Center by a trained researcher using a five-step automated multi-pass method called the USDA Automated Multiple-Pass Method [20]. The USDA Food and Nutrient Database for Dietary Studies (FNDDS) 2017–2018 was used to define the food categories within the NHANES 2017–2018 dietary intake data [21]. The FNDDS 2017–2018 contains approximately 7100 food and beverage items [21]. Data on food items for the 12 food groups regarding the three Japanese diet indices were used in the present analysis. Details of the definitions of these 12 food groups are provided in the Appendix A.

### 2.4. Japanese Diet Indices

We defined the three types of Japanese diet indices used in previous studies [12,13].

First, we defined JDI from nine food items: (1) rice, (2) miso soup, (3) fish, (4) green and yellow vegetables, (5) seaweed, (6) pickles, (7) green tea, (8) beef and pork, and (9) coffee. For these nine food items of the JDI, seven adhering components (“rice”, “miso soup”, “fish”, “green and yellow vegetables”, “seaweed”, “pickles”, and “green tea”) and two non-adhering components (“beef and pork” and “coffee”) were considered as characters of the Japanese diet. For each of the seven adhering components, participants were assigned one point if their daily intake was equal to or greater than the sex-specific median and zero points if their daily intake was below the sex-specific median. For each of the two non-adhering components (“beef and pork” and “coffee”), participants were assigned one point if their daily intake was below the sex-specific median and zero point if their daily intake was equal to or greater than the sex-specific median. Thus, the JDI scores ranged from 0 to 9. Higher JDI scores indicate greater conformity to the Japanese diet.

Second, mJDI was defined from 12 food items, and three adhering components (“soybeans and soybean foods”, “fruits”, and “mushrooms”) were added to the JDI. Thus, the mJDI had a 12-point scale (i.e., 10 adhering items and two non-adhering items).

Third, we defined the wJDI, which comprised selected components that were significantly associated with the nutrient density score from the 12 food items of the mJDI. First, we selected food items that were significantly associated with the nutrient density score using a multiple linear regression model, after which weighted points were calculated using the standardized regression coefficients obtained from the multiple linear regression model. Specifically, the weighted points of each selected food item were calculated as the standardized regression coefficient multiplied by 10 (for example, if the standardized regression coefficient was “0.1”, the weighted point of this food item was “+1 point”). Finally, we added weighted points to each selected food item based on the sex-specific median.

### 2.5. Nutrient Density Score

The nutrient density score was calculated based on the Nutrient-Rich Food Index 9.3 (NRF 9.3) [22,23,24]. The NRF 9.3 consists of nine nutritional components to encourage (protein, dietary fiber, iron, vitamin A, vitamin C, vitamin E, calcium, potassium, and magnesium) and three nutritional components to limit (sodium, saturated fatty acids, and sugar) [22,23,24]. The nutrient density of each nutritional component was defined as the percentage of sufficiency relative to the daily reference values for the United States population (Table 1).

Considering that the intake of various nutrients is also high when the overall intake volume is high, we used the density method as an adjustment for total energy intake, and each intake value of the nutritional component was standardized to 2000 kcal (because 2000 kcal is an Estimated Average Requirement in the US). In addition, the nutrient density of each component was capped at 100% to avoid abnormally high values due to outliers, as in a previous study [22]. The nutrient density score was calculated using the following formula:Nutrient Density Score = ∑ 9 encouraged nutrient densities − ∑ 3 limit nutrient densities

Higher nutrient density scores indicate higher overall nutrient sufficiency.

### 2.6. Covariates

In the present study, race was used for the sensitivity analysis (stratified analysis). In the NHANES, the variable data of race consisted of the following categories: “Mexican American”, “Other Hispanic”, “Non-Hispanic White”, “Non-Hispanic Black”, “Non-Hispanic Asian”, and “Other Race—Including Multi-Racial”.

### 2.7. Statistical Analysis

Spearman’s rank correlation coefficients were calculated for the correlation analysis between the three Japanese diet indices and the nutrient density score. Tests for differences in the correlation coefficients for each of these three Japanese diet indices were also performed using R “cocor” package.

For the sensitivity analysis, the correlation coefficients between the three Japanese diet indices and the 12 nutritional components comprising the nutrient density score were calculated. In addition, the correlation coefficients between the three Japanese diet indices and the sodium/potassium ratio were calculated.

We performed two sensitivity analyses. First, energy adjustments were applied to each food item in the three Japanese diet indices using the residual method, and the median value was calculated using these energy-adjusted variables to define the three Japanese diet indices [25]. Second, we conducted a stratified analysis by race on the correlation between the three Japanese diet indices and the nutrient density score.

Statistical significance was set at *p* < 0.05. To define the wJDI, we selected food items from the mJDI that were statistically significant (*p* < 0.05). All statistical analyses were performed using R ver. 4.2.2. In addition, the “cocor” package was used to test for differences in the correlation coefficients “http://comparingcorrelations.org/ (accessed on 22 March 2024)”.

### 2.8. Ethical Considerations

NHANES was conducted with the approval of the National Center for Health Statistics Research Ethics Review Board [26]. The present study was based only on the published NHANES data “https://wwwn.cdc.gov/nchs/nhanes/ (accessed on 22 March 2024)”.

## 3. Results

### 3.1. Basic Characteristics

Of the 3138 participants, 1462 (46.6%) were male and 1676 (53.4%) were female, with a mean age (standard deviation) of 50.4 (16.2) years. The racial distribution was as follows: 438 people (14.0%) were Mexican American, 305 people (9.7%) were Other Hispanic, 1086 people (34.6%) were Non-Hispanic White, 719 people (22.9%) were Non-Hispanic Black, 436 people (13.9%) were Hispanic Asian, and 154 people (4.9%) were of other races, including those who were multi-racial.

The distributions of the JDI and mJDI are shown in Figure 2 and Figure 3**,** and corresponding frequency distribution tables are also shown in the Appendix A. The mean (standard deviation) of JDI was 2.0 (1.1) points, and the mean (standard deviation) of mJDI was 2.8 (1.4) points.

### 3.2. Correlation between JDI/mJDI and Nutrient Density

Figure 4a shows the correlation between JDI and nutrient density score. The correlation coefficient between the JDI and the nutrient density score was 0.24 (*p* < 0.001).

Figure 4b shows the correlation between the mJDI and nutrient density score. The correlation coefficient between the mJDI and the nutrient density score was 0.38 (*p* < 0.001). There was a statistically significant difference between the correlation coefficients of JDI and mJDI (*p* < 0.001 for the difference in correlation coefficients).

### 3.3. Food Items of mJDI and Nutrient Density

The associations between each food item in the mJDI and the nutrient density scores are shown in Table 2. Six food items were statistically significantly associated with the nutrient density score: “green and yellow vegetables”, “green tea”, “soybeans and soybean foods”, “fruits”, “mushrooms”, and “beef and pork”. The standardized coefficients of these six food items were 0.3 for “green and yellow vegetables”, 0.0 for “green tea”, 0.2 for “soybeans and soybean foods”, 0.3 for “fruits”, 0.0 for “mushrooms”, and 0.1 for “beef and pork”. Therefore, the wJDI added +3 points for “green and yellow vegetables”, +2 points for “soybeans and soybean foods”, and +3 points for “fruits” when their daily intake was equal to or over the sex-specific median and added +1 point for “beef and pork” when their daily intake was below the sex-specific median. The wJDI scores ranged from 0 to 9 points. The wJDI distribution is shown in Figure 5, and the corresponding frequency distribution table is shown in the Appendix A. The mean (standard deviation) wJDI was 4.1 (2.5).

### 3.4. Correlation between wJDI and Nutrient Density

Figure 6 shows the correlation between the wJDI and the nutrient density score. The correlation coefficient between the wJDI and the nutrient density score was 0.48 (*p* < 0.001), which was statistically significantly higher than that between the JDI (*p* < 0.001) and mJDI (*p* < 0.001).

### 3.5. Correlation between Japanese Diet Indices and Nutritional Components

The correlations between the three Japanese diet indices and each nutritional component comprising the nutrient density scores are shown in Table 3. The JDI was positively correlated with all nine nutritional components to encourage, although calcium was not statistically significant. The mJDI and wJDI scores were statistically significantly and positively correlated with all nine components. Regarding the three nutritional components to limit, the JDI was statistically significantly correlated with all three components, negatively correlated with saturated fatty acids and sugar, and positively correlated with sodium. The mJDI was significantly correlated with sodium and saturated fatty acids, negatively correlated with saturated fatty acids, and positively correlated with sodium levels. The wJDI was statistically significantly correlated with all three components, negatively correlated with saturated fatty acids, and positively correlated with sodium and sugar.

Correlations between the three energy-adjusted Japanese diet indices and the nutrient density score obtained using the residual method were similar to the aforementioned results, except that the correlation coefficient for calcium in the JDI was relatively higher.

JDI was positively correlated with the sodium/potassium ratio, whereas mJDI and wJDI were negatively correlated. All three energy-adjusted Japanese diet indices negatively correlated with the sodium/potassium ratio.

### 3.6. Stratified Analysis by Race

The correlations between the three Japanese diet indices and nutrient density scores stratified by race are shown in Table 4. A positive correlation was found between the three Japanese diet indices and nutrient density scores in all racial groups.

## 4. Discussion

In the present study, we conducted cross-sectional analyses using the NHANES data to examine the association between three Japanese diet indices and nutrient density scores in the United States population. The results of the present study suggest that higher Japanese diet indices (JDI, mJDI, and wJDI) were associated with higher nutrient densities. In addition, the mJDI and wJDI were more strongly correlated with the nutrient density score than the JDI. These results are essentially similar to those of a previous study in a Japanese population [12].

The present study showed that the three Japanese diet indices and each nutritional component were positively correlated with all nine nutritional components. These results suggest that the Japanese diet was associated with better nutrient intake. In contrast, all three Japanese diet indices were positively correlated with sodium intake, similar to a previous study on the Japanese population [12]. However, it is unlikely that the food items of the Japanese diet indices are the major sodium food sources among the United States population since the top 10 sodium food sources in the United States population (NHANES 2017–2018) are reported to be pizza; breads, rolls, and buns; cold cuts and cured meats; soups; burritos and tacos; savory snacks; poultry; cheese; pasta mixed dishes; and burgers [27]. As sodium intake is a risk factor for several chronic diseases, such as hypertension and gastric cancer [28,29,30,31,32,33], high sodium intake may be a disadvantage of the diet defined by the Japanese diet indices in the present study. However, the correlation with the sodium/potassium ratio was negative for both mJDI and wJDI. A high risk of cardiovascular outcomes from cardiovascular has been reported in individuals with high dietary sodium/potassium ratios [34,35]. Therefore, the disadvantages of high sodium intake may be offset by high potassium intake in the mJDI and wJDI groups.

Similar to previous studies in the Japanese population [12], the results of the present study showed that the mJDI and wJDI were more strongly associated with the nutrient density score than the JDI. However, the correlation coefficients were lower in the present study than in previous studies of the Japanese population; the correlation coefficients were 0.34 for the JDI, 0.44 for the mJDI, and 0.61 for the wJDI in the previous studies of the Japanese population [12], whereas, in the present study, they were 0.24 for the JDI, 0.38 for the mJDI, and 0.48 for the wJDI as shown in Figure 4 and Figure 6. One reason for this difference may be the lower intake volume of food items in the Japanese diet; the proportion of intake exceeding 0 g was <10% for five food items (miso soup, seaweed, pickles, green tea, and mushrooms) in the present study. Because the three Japanese diet indices were calculated based on the sex-specific median in the study population, it is assumed that the Japanese diet indices were added despite fewer dietary intakes of the Japanese diet than in a previous study on the Japanese population.

Furthermore, the food items selected as the wJDI in the present study differed from those used in previous studies on the Japanese population [12]. In a previous study on the Japanese population, nine food items (“rice”, “fish and shellfish”, “green and yellow vegetables”, “seaweed”, “green tea”, “soybeans and soy products”, “fruits”, “mushrooms”, and “beef and pork”) were identified as food items of the wJDI [12]. Conversely, in the present study, just four food items were selected: “green and yellow vegetables”, “soybeans and soybean foods”, “fruits”, and “beef and pork”. The reason for this result in the wJDI may be that food intake related to the Japanese diet was lower. However, the result that the weights of “green and yellow vegetables”, “soybeans and soybean foods”, and “fruits” were relatively higher was consistent with previous studies of the Japanese population [12]. These findings indicate that the JDI and mJDI can be improved in terms of nutrient intake to be generalizable to any region by revising or weighing these three food items.

In a previous study examining the association between race and diet quality (the Healthy Eating Index) in the United States population, it was reported that the Healthy Eating Index differed by race and that there were differences in food intake, including vegetables, fruits, and meat [36]. Therefore, we also conducted analyses stratified by race to examine whether there was a positive correlation between the Japanese diet indices and nutrient density score, even when non-Japanese (non-Asian) individuals were analyzed. The results showed that all three Japanese diet indices were positively correlated with nutrient density scores in all racial categories (Table 4). Therefore, these Japanese diet indices may be useful as dietary information in terms of nutrient density, not only among Asians but also among various other racial groups.

The present study had several limitations. First, we used dietary assessment data from the 24 h recall method in the present study. Because the 24 h recall method depends on the participant’s memory, it is undeniable that there were misclassifications due to omissions in reporting [37,38,39]. Second, we may not have selected all food items in the FNDDS to qualify for the Japanese diet indices. For example, “Mixed Dish”, a WWEIA food category that combines multiple foods (e.g., “dirty rice” would contain a certain amount of rice), might include food items of the Japanese diet indices, although the weight of each food item of the Japanese diet indices cannot be ascertained. As we did not include “Mixed Dish” in the Japanese diet indices in the present study, we cannot rule out the possibility that this incurred bias. Third, the NRF 9.3 Index, which is used as an indicator of nutrient density, was limited to 12 nutritional components, and we did not examine other nutritional factors that are particularly expected in the Japanese diet. For example, manganese, intestinal bacteria, and metabolites derived from intestinal bacteria have been reported to be advantages of the Japanese diet [40,41,42].

## 5. Conclusions

Even in the United States, which includes various non-Asian racial groups, correlations between Japanese diet indices (JDI and mJDI) and nutrient density were replicated. Therefore, the results of the present study suggest that the application of these indices to non-Japanese populations may be useful for examining desirable dietary patterns in terms of nutrient intake. In addition, because the mJDI and wJDI were more strongly correlated with the nutrient density score than the JDI, they may be more useful in terms of nutrient intake. This result implies that it may be possible to improve nutrient intake not only in the population of Japan but also in the United States and other Western countries by recommending a Japanese diet based on the Japanese diet indices as a public health nutrition measure.

However, because the three Japanese diet indices were correlated with high sodium intake, high sodium intake may be a major issue in the Japanese diet. Further studies are warranted to determine the ideal definition of Japanese dietary patterns that can be accepted internationally.

## Figures and Tables

**Figure 1 nutrients-16-02431-f001:**
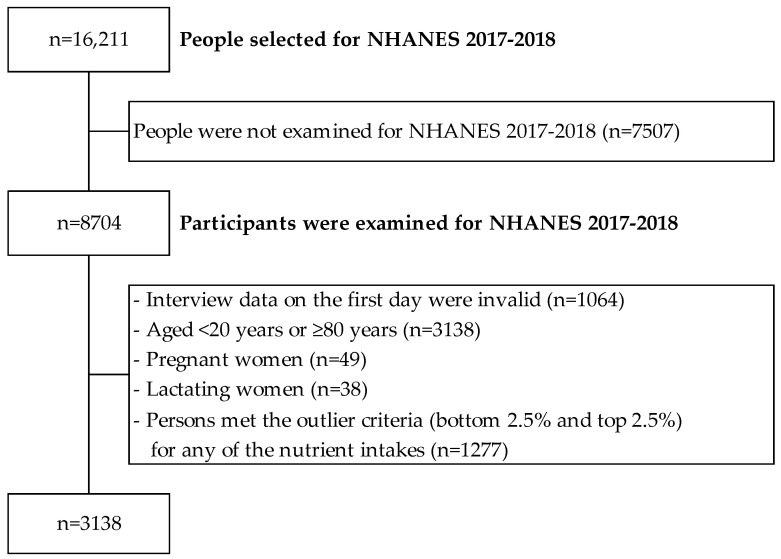
Flow chart of research participants.

**Figure 2 nutrients-16-02431-f002:**
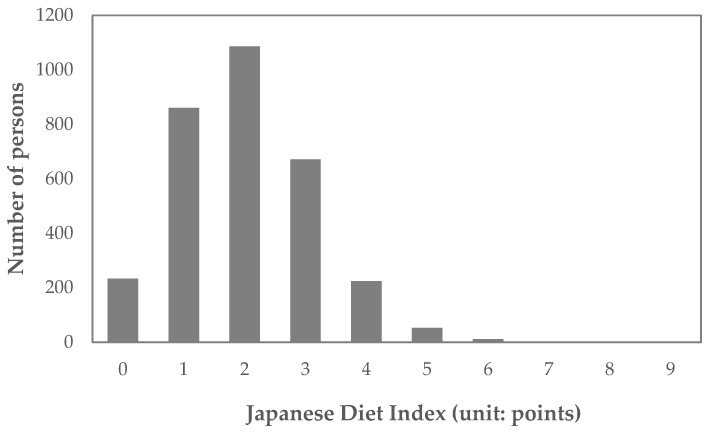
Distribution of the Japanese Diet Index (JDI).

**Figure 3 nutrients-16-02431-f003:**
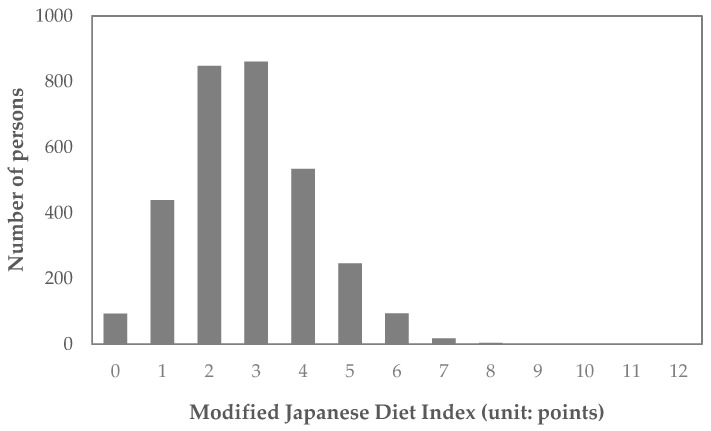
Distribution of the modified Japanese Diet Index (mJDI).

**Figure 4 nutrients-16-02431-f004:**
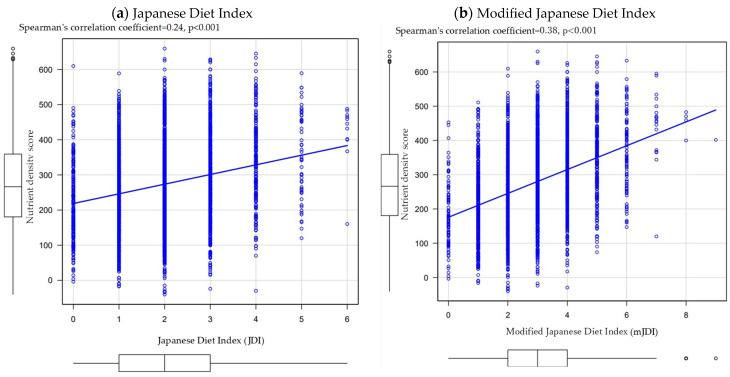
Correlation between the Japanese diet indices and nutrient density. (**a**) Correlation between the Japanese Diet Index (JDI) and nutrient density. (**b**) Correlation between the modified Japanese Diet Index (mJDI) and nutrient density.

**Figure 5 nutrients-16-02431-f005:**
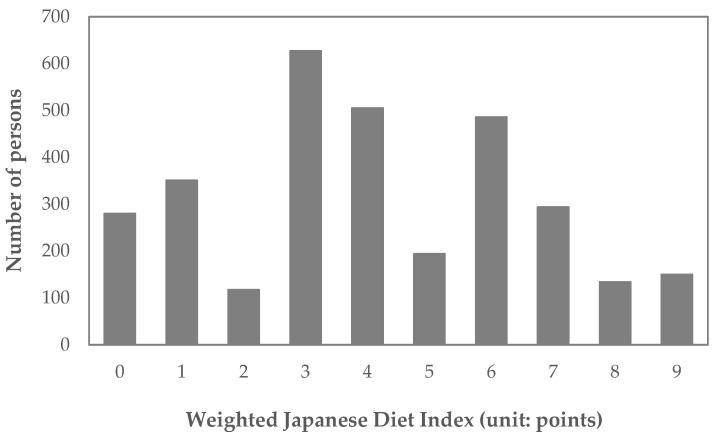
Distribution of the weighted Japanese Diet Index (wJDI).

**Figure 6 nutrients-16-02431-f006:**
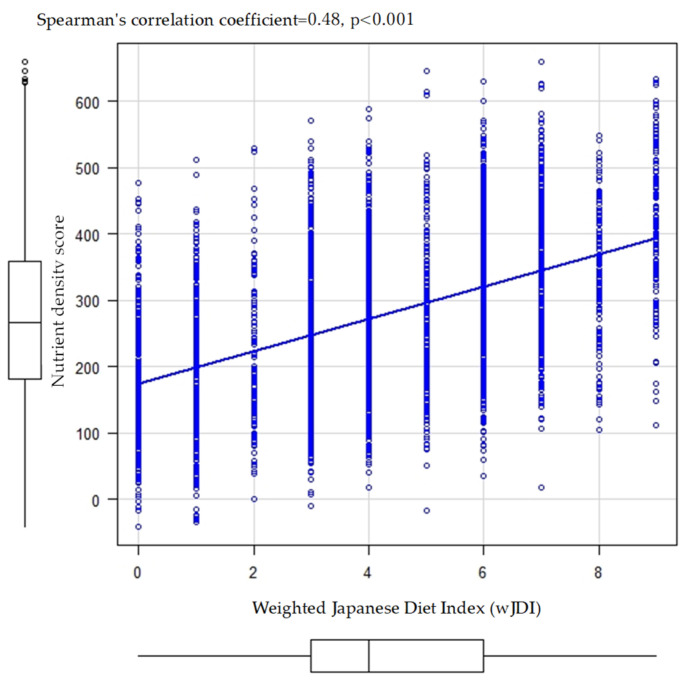
Correlation between the weighted Japanese Diet Index (wJDI) and nutrient density.

**Table 1 nutrients-16-02431-t001:** Reference intake values for each nutritional component for nutrient density score.

	Reference Intake Values ^1^
Protein (g)	50
Dietary fiber (g)	28
Iron (g)	18
Vitamin A (µg)	900
Vitamin C (µg)	90
Vitamin E (µg)	20
Calcium (mg)	1300
Potassium (mg)	4700
Magnesium (mg)	420
Sodium (mg)	2300
Saturated fatty acids (g)	20
Sugar (g)	50

^1^ Dietary reference intakes.

**Table 2 nutrients-16-02431-t002:** Association between mJDI food items and nutrient density.

Explanatory Variables ^1^	Beta ^2^	Standard Error ^3^	*p*	Standardized Beta ^4^
Rice	8.5	4.8	0.078	-
Miso soup	17.5	24.8	0.480	-
Fish	8.7	4.6	0.057	-
Green and yellow vegetables	77.5	3.5	<0.001	0.3
Seaweed	−1.2	30.8	0.968	-
Pickles	8.0	6.2	0.192	-
Green tea	14.5	6.7	0.031	0.0
Soybeans and soybean foods	41.5	3.8	<0.001	0.2
Fruits	76.0	3.5	<0.001	0.3
Mushrooms	31.8	12.3	0.010	0.0
Beef and pork	22.1	3.5	<0.001	0.1
Coffee	−6.5	3.4	0.060	-
Energy intake (kcal)	−0.07	0.002	<0.001	

^1^ Food items of modified Japanese Diet Index and energy intake. ^2^ Unstandardized regression coefficients for each food item with 0 or 1 point (above median) as the explanatory variable. ^3^ Standard error of unstandardized regression coefficients. ^4^ Standardized regression coefficients. Standardized beta listed only for food items that were statistically significant (*p* < 0.05).

**Table 3 nutrients-16-02431-t003:** Correlation between the Japanese diet indices and each component of the nutrient density score.

	Crude						Energy-Adjusted					
	JDI ^1^		mJDI ^2^		wJDI ^3^		JDI ^1^		mJDI ^2^		wJDI ^3^	
	CC ^4^	*p*	CC ^4^	*p*	CC ^4^	*p*	CC ^4^	*p*	CC ^4^	*p*	CC ^4^	*p*
Energy (kcal)	−0.02	0.226	−0.01	0.470	0.03	0.062						
Protein (g)	0.06	<0.001	0.07	<0.001	0.05	0.006	0.08	<0.001	0.11	<0.001	0.09	<0.001
Dietary fiber (g)	0.22	<0.001	0.43	<0.001	0.54	<0.001	0.26	<0.001	0.41	<0.001	0.55	<0.001
Iron (g)	0.06	0.001	0.14	<0.001	0.18	<0.001	0.15	<0.001	0.22	<0.001	0.23	<0.001
Vitamin A (µg)	0.18	<0.001	0.21	<0.001	0.31	<0.001	0.22	<0.001	0.26	<0.001	0.34	<0.001
Vitamin C (µg)	0.24	<0.001	0.35	<0.001	0.41	<0.001	0.24	<0.001	0.33	<0.001	0.44	<0.001
Vitamin E (µg)	0.20	<0.001	0.24	<0.001	0.26	<0.001	0.17	<0.001	0.19	<0.001	0.26	<0.001
Calcium (mg)	0.01	0.614	0.05	0.008	0.11	<0.001	0.10	<0.001	0.14	<0.001	0.14	<0.001
Potassium (mg)	0.11	<0.001	0.25	<0.001	0.36	<0.001	0.21	<0.001	0.34	<0.001	0.43	<0.001
Magnesium (mg)	0.17	<0.001	0.31	<0.001	0.37	<0.001	0.27	<0.001	0.40	<0.001	0.43	<0.001
Sodium (mg)	0.17	<0.001	0.11	<0.001	0.04	0.024	0.15	<0.001	0.11	<0.001	0.06	<0.001
Saturated fatty acids (g)	−0.20	<0.001	−0.23	<0.001	−0.16	<0.001	−0.21	<0.001	−0.24	<0.001	−0.19	<0.001
Sugar (g)	−0.05	0.002	0.00	0.823	0.04	0.017	0.00	0.876	0.03	0.067	0.05	<0.001
Sodium/potassium	0.04	0.034	−0.12	<0.001	−0.26	<0.001	−0.05	0.004	−0.18	<0.001	−0.30	<0.001

^1^ Japanese Diet Index (9 items). ^2^ Modified Japanese Diet Index (12 items). ^3^ Weighted Japanese Diet Index (6 items selecting and weighting from mJDI food items). ^4^ Spearman’s rank correlation coefficients.

**Table 4 nutrients-16-02431-t004:** Association between Japanese diet indices and nutrient density: stratified analysis by race.

	JDI ^1^		mJDI ^2^		wJDI ^3^	
	CC ^4^	*p*	CC ^4^	*p*	CC ^4^	*p*
Mexican American	0.15	<0.001	0.33	<0.001	0.44	<0.001
Other Hispanic	0.33	<0.001	0.49	<0.001	0.55	<0.001
Non-Hispanic White	0.19	<0.001	0.33	<0.001	0.44	<0.001
Non-Hispanic Black	0.26	<0.001	0.37	<0.001	0.44	<0.001
Non-Hispanic Asian	0.18	<0.001	0.28	<0.001	0.42	<0.001
Other Races—Including Multi-Racial	0.21	0.009	0.35	<0.001	0.44	<0.001

^1^ Japanese Diet Index (without energy adjustment). ^2^ Modified Japanese Diet Index (without energy adjustment). ^3^ Weighted Japanese Diet Index (without energy adjustment). ^4^ Spearman’s rank correlation coefficients.

## Data Availability

NHANES data are publicly available via the website “https://wwwn.cdc.gov/nchs/nhanes/ (accessed on 22 March 2024)”.

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
