# Peer review of "Japanese Diet Indices and Nutrient Density in US Adults: A Cross-Sectional Analysis with NHANES Data"

_nutrients, 2024, doi:10.3390/nu16152431_

Round 1

Reviewer 1 Report

Comments and Suggestions for Authors

The manuscript entilted "Japanese diet indices and nutrient density in US adults: A cross-sectional analysis with NHANES data". Several problems should be revised as follows:

1. Please add the inclusion criteria for participants,ffor example, which age group was included? 

2. Why did the authors exclude the participants who  were potential outliers in the variables of nutrient 90 intake used in the present analysis (bottom 2.5% and top 2.5%; total of 5%)? 

3. If there is also a significant relationship between Japanese diet indices (JDI and mJDI) and nutrient density in other populations who intaked Japanese diet? And what are the applications of your findings?

4. The manscript content format should follow the journal requirements.

5. All abbreviations in the Tables should be noted below the Tables. For example, CC.

Author Response

RESPONSE TO REVIEWER 1:

We sincerely appreciate the reviewers’ efforts in carefully reading our manuscript. In response to your comment, we have revised the manuscript – please see the responses below.

Comment 1:

Please add the inclusion criteria for participants, for example, which age group was included?

Response: We added the following text to the manuscript to address these points:

Line 84: The NHANES target population is the noninstitutionalized civilian resident population of the United States [17,18]. The NHANES 2017–2018 survey included all ages [15-17].

Comment 2:

Why did the authors exclude the participants who were potential outliers in the variables of nutrient intake used in the present analysis (bottom 2.5% and top 2.5%; total of 5%)?

Response: We added the following text to the manuscript to address this point:

Line 90: Furthermore, among 4,415 people, we also excluded 1,277 people who were potential outliers in the variables of nutrient intake used in the present analysis (bottom 2.5% and top 2.5%; total of 5%). Since the distribution of energy intake is widely spread from 0 to 12,501 kcal in NHANES 2017–2018 survey data, previous studies have reported the need to consider outliers focusing on numeric values of nutrient intake [19].

Comment 3:

If there is also a significant relationship between Japanese diet indices (JDI and mJDI) and nutrient density in other populations who intaked Japanese diet? And what are the applications of your findings?

Response: We added the following text to the manuscript to address this point:

Line 355: This result implies that it may be possible to improve nutrient intake not only population in Japan but also in the United States and other Western countries by recommending a Japanese diet based on the Japanese diet indices as a public health nutrition measure.

However, because the three Japanese diet indices correlated with high sodium intake, high sodium intake may be a major issue in the Japanese diet. Further studies would be warranted to determine the ideal definition of Japanese dietary pattern that can be accepted internationally.

Comment 4:

The manscript content format should follow the journal requirements.

All abbreviations in the Tables should be noted below the Tables. For example, CC.

Response: According to this comment, we revised Tables such as Table 4.

Reviewer 2 Report

Comments and Suggestions for Authors

The authors describe the distribution of the Japanese Diet Index (JDI), the Modified Japanese Diet Index (mJDI), and the weighted Japanese Diet Index (wJDI) in Figures 2, 3, and 6, respectively. I recommend replacing these figures with a descriptive table that provides a comprehensive overview of all three Japanese diet indices.

I recommend combining Figures 4, 5, and 6 into a single Figure 4, with sections a, b, and c, providing a comprehensive overview of Correlation between all three Japanese diet indices and nutrient density.

Author Response

RESPONSE TO REVIEWER 2:
The authors describe the distribution of the Japanese Diet Index (JDI), the Modified Japanese Diet Index (mJDI), and the weighted Japanese Diet Index (wJDI) in Figures 2, 3, and 6, respectively. I recommend replacing these figures with a descriptive table that provides a comprehensive overview of all three Japanese diet indices.

I recommend combining Figures 4, 5, and 6 into a single Figure 4, with sections a, b, and c, providing a comprehensive overview of Correlation between all three Japanese diet indices and nutrient density.

Response: We sincerely appreciate the reviewers’ efforts in carefully reading our manuscript. In response to your comment, we have revised the figures.

Reviewer 3 Report

Comments and Suggestions for Authors

The study by Aono et al entitled "Japanese diet indices and nutrient density in US adults: A cross-sectional analysis with NHANES data" was to examine the association between three Japanese diet indices and nutrient density among populations of various ethnicities in the United States.

I found the article interesting. However, his organization should be reviewed. The authors include numerical values ​​of the correlation coefficients in the discussion, but not in the results. I suggest doing the opposite (L294-296).

Furthermore, I have a curiosity: how did the authors, affiliated with a Japanese structure, obtain the data used for the study from NHANES?

The authors report that “Dietary survey data were based on the 24-h recall method conducted at the Mobile Examination Center by a trained researcher”: were they provided by NHANES or self-obtained? If so, how did they track down the 3138 participants?

Author Response

RESPONSE TO REVIEWER 3:
The study by Aono et al entitled "Japanese diet indices and nutrient density in US adults: A cross-sectional analysis with NHANES data" was to examine the association between three Japanese diet indices and nutrient density among populations of various ethnicities in the United States.

Comment 1:
I found the article interesting. However, his organization should be reviewed. The authors include numerical values of the correlation coefficients in the discussion, but not in the results. I suggest doing the opposite (L294-296).

Response: We added the following text to the manuscript to address this point:

Line 193: Figure 4.a shows the correlation between JDI and nutrient density score. The correlation coefficient between the JDI and the nutrient density score was 0.24 (p<0.001).
Figure 4.b shows the correlation between the mJDI and nutrient density score. The correlation coefficient between the mJDI and the nutrient density score was 0.38 (p<0.001).

Line 304: as shown in Figure 4 and Figure 6.

Comment 2:
Furthermore, I have a curiosity: how did the authors, affiliated with a Japanese structure, obtain the data used for the study from NHANES? The authors report that “Dietary survey data were based on the 24-h recall method conducted at the Mobile Examination Center by a trained researcher”: were they provided by NHANES or self-obtained? If so, how did they track down the 3138 participants?

Response: As shown in Data Availability Statement (line 383), NHANES data is publicly available via the website (https://wwwn.cdc.gov/nchs/nhanes/). Therefore, you can access all of data which we used (i.e. published anonymized data). If you are interested in specific procedures of the present study, please contact the correspondence author (Email: [email protected]).